# Survival outcomes of surgery for retroperitoneal sarcomas: A systematic review and meta-analysis

**Qiang Guo, Jichun Zhao, Xiaojiong Du\*, Bin Huang** \*

Department of Vascular Surgery, West China Hospital, Sichuan University, Chengdu, Sichuan Province, China

\* duxiaojiong@163.com (XD); bhwchscu@163.com (BH)

## Abstract

### Background

Definitive evidence to guide clinical practice on the principles of surgery for retroperitoneal sarcomas (RPSs) is still lacking. This study aims to summarise the available evidence to assess the relative benefits and disadvantages of an aggressive surgical approach with contiguous organ resection in patients with RPS, the association between surgical resection margins and survival outcomes, and the role of surgery in recurrent RPS.

### Methods

We searched PubMed, the Cochrane Library, and EMBASE for relevant randomised trials and observational studies published from inception up to May 1, 2021. Prospective or retrospective studies, published in the English language, providing outcome data with surgical treatment in patients with RPS were selected. The primary outcome was overall survival (OS).

### Findings

In total, 47 articles were analysed. There were no significant differences in the rates of OS (HR: 0.93; 95% CI: 0.83–1.03; $P$ = 0.574) and recurrence-free survival (HR: 1.00; 95% CI: 0.74–1.27; $P$ = 0.945) between the extended resection group and the tumour resection alone group. Organ resection did not increase postoperative mortality (OR: 1.00; 95% CI: 0.55–1.81; $P$ = 0.997) but had a relatively higher complication rate (OR: 2.24, 95% CI: 0.94–5.34; $P$ = 0.068). OS was higher in R0 than in R1 resection (HR: 1.34; 95% CI: 1.23–1.44; $P$ < 0.001) and in R1 resection than in R2 resection (HR: 1.86; 95% CI: 1.35–2.36; $P$ < 0.001). OS was also higher in R2 resection than in no surgery (HR: 1.26; 95% CI: 1.07–1.45; $P$ < 0.001), however, subgroup analysis showed that the pooled HR in the trials reporting primary RPS was similar between the two groups (HR, 1.14; 95% CI, 0.87–1.42; $P$ = 0.42). Surgical treatment achieves a significantly higher OS rate than does conservative treatment (HR: 2.42; 95% CI: 1.21–3.64; $P$ < 0.001) for recurrent RPS.

**Data Availability Statement:** All relevant data are within the paper and its Supporting Information files.

**Funding:** The authors received no specific funding for this work.

**Competing interests:** The authors have declared that no competing interests exist.

## Conclusions

For primary RPS, curative-intent en bloc resection should be aimed, and adjacent organs with evidence of direct invasion must be resected to avoid R2 resection. For recurrent RPS, surgical resection should be considered as a priority. Incomplete resection remains to have a survival benefit in select patients with unresectable recurrent RPS.

## Introduction

Soft tissue sarcomas (STS) are rare malignant tumours that most commonly arise from cells of mesenchymal origin and represents approximately 1% of all adult malignancies [1]. Approximately 15–20% of all STSs arise in the retroperitoneum [2]. STS consists of more than 70 well-defined histologic subtypes, and liposarcoma is the most common one found in the retroperitoneum [2]. Other subtypes include leiomyosarcoma, MFH, solitary fibrous tumors and malignant peripheral nerve sheath tumors [2]. Individual histologic subtypes have unique behavioral characteristics and treatment outcomes. Although STS of the retroperitoneum are rare, these tumours have worse prognosis than those arising from the trunk or extremity, with 5-year overall survival (OS) rates of 39–70% [3]. Several factors influence this poor prognosis. First, retroperitoneal sarcomas often progress asymptomatically and are thus only detected incidentally when the substantially enlarged tumour compresses the surrounding organs [4]. Patients presenting with back pain or abdominal distention already have a large tumour with multi-organ involvement and close proximity to critical structures such as major vessels or kidney. Second, surgical resection of localised tumours with gross negative margins remains the mainstay of curative treatment for patients with primary retroperitoneal sarcomas (RPSs) [5]. However, a significant percentage of patients, even those treated at high-volume centres with gross negative margins, develop disease recurrence [6]. Besides, recent multicentre randomised controlled trials (RCTs) have reported similar rates of abdominal recurrence-free survival (RFS) and OS between surgery alone and preoperative radiotherapy plus surgery [7]. Adjuvant chemotherapy is not routinely recommended in RPS because of lack of sufficient evidence supporting its OS benefit [3]. Third, RPS has over 70 different histologic subtypes, and the heterogeneity in its biological behaviour, treatment response, and oncological risks renders a homogeneous therapeutic approach difficult and makes it challenging to develop evidence-based guidelines [8].

Surgery for primary or recurrent RPS is still technically challenging [5]. Thus, margin assessment continues to be an area of uncertainty in RPS surgery. Actual pathologic evidence of organ invasion is rare, and thus, the appropriateness of resecting adjacent uninvolved organs in RPS surgeries is still controversial [9]. Aggressive resection to grossly uninvolved organs may improve R0 resection rates; however, the benefit of converting R1 to R0 resections is unclear, and concomitant organ resection might be associated with an increased risk of postoperative complications [10]. Currently, local recurrence is the primary cause of mortality in RPS, with up to 75% of mortalities occurring without evidence of distant metastases [11]. Although R2 resection is not recommended for primary RPS, some study suggested that R2 resection may prolong survival and alleviate symptoms in select patients with unresectable RPS [12]. Further, data regarding the outcomes of surgery for recurrent RPS and data to guide treatment decisions for patients with local recurrence are limited.

Thus, we aimed to gather available evidence to determine the relative benefit and disadvantages of an aggressive surgical approach with contiguous organ resection in patients with RPS.

We also compared the long-term survival rates among different surgical resection margins for RPS and the OS rates between surgery and conservative treatment in patients with recurrent RPS.

## Methods

### Search strategy

This systematic review and meta-analysis was conducted according to the Preferred Reporting Items for Systematic Reviews and Meta-analyses (PRISMA) guidelines for systematic reviews [13]. We searched PubMed, the Cochrane Library, and EMBASE for relevant studies published from inception up to May 1, 2021 using the following keywords: 'retroperitoneal tumour' or 'retroperitoneal neoplasm' or 'retroperitoneal sarcoma' and 'surgery' or 'surgical' or 'resection' (specific search strategies are listed in S1 File).

We included randomized trials and observational studies comparing different surgical resection margins for RPS; comparing surgery with conservative treatment for recurrent RPS patients; and comparing extended resection including adjacent organs with resection of tumour alone. Conference abstracts, letters, editorials, or any publication other than a peer-reviewed original research article or a technical report from a national public health organization and those that did not provide hazard ratios (HRs) or confidence intervals (CIs) were excluded. Studies were also excluded if the study population was duplicated in another study included in our meta-analysis. In case of duplicate populations, the study that included more institutions or more patients was selected. Only studies published in the English language were included, and the references of the selected articles were reviewed for additional relevant studies.

### Data extraction and quality assessment

Two authors (Q.G. and X.D.) independently selected the studies based on the inclusion and exclusion criteria. After the initial search, the titles and abstracts were independently screened to identify potentially relevant studies that were then submitted to a full-text review. Disagreements were resolved by discussion with a third reviewer (J.Z.). The following data were compiled in a spreadsheet: (1) study characteristics (name of the first author, publishing year, study design, sample size); (2) tumour characteristics (histologic subtype, French Federation of Cancer Centers Sarcoma Group [FNCLCC] grade [14], tumour status); (3) surgical characteristics (combined organ resection, margin status, vascular reconstruction); (4) adjuvant therapy (radiotherapy/chemotherapy), and (5) outcomes (OS, RFS, postoperative complications, and 30-day mortality). When data were unavailable, efforts were made to contact the corresponding author to obtain the missing data.

The methodological quality of the studies was assessed using the Newcastle-Ottawa Scale [15]. The scale evaluates study bias and assigns points in the following three domains: patient selection, comparability, and outcomes. Each reviewer generated a score, and the value was reviewed (Q.G. and J.Z.). Studies with a high risk of bias (score <6) were further reviewed for inclusion.

### Statistical analysis

All outcomes were dichotomous data. Heterogeneity was assessed using the $I^2$ statistic, with $I^2$ values of 25%, 50%, and 75% considered to indicate low, moderate, and high heterogeneity, respectively. The primary outcome was OS. The secondary outcomes were RFS, postoperative complications and early postoperative mortality. Pooled HRs and 95% CIs were estimated to

compare the risk of recurrence or OS. Pooled odds ratios (ORs) with 95% CIs were estimated to compare the risk of postoperative complication or early postoperative mortality between an aggressive surgical approach with contiguous organ resection and tumour resection alone. For time-to-event outcomes, including RFS and OS, HRs and their associated variances were extracted, or estimates were calculated where possible using the methods described by Tierney et al [16]. Prespecified subgroup analyses by tumour status (primary/recurrent) were performed. Sensitive analyses that only including studies with similar surgical margins were also performed. Publication bias was assessed using funnel plots. All statistical analyses were performed using Stata/MP, version 16.0 (StataCorp LLC). All tests were two sided, and P<0.05 was considered statistically significant.

## Results

### Study characteristics

A total of 4172 articles were initially evaluated, and 16 studies were further identified through the references. After removing the 1384 duplicates, the titles and abstracts of 2804 articles were reviewed. Among them, 238 studies were reviewed in full text. Finally, 47 studies involving 22608 patients were included in the final analysis [10, 12, 17–61]. All 47 studies were observational research. The PRISMA flow diagram showing the entire review process from the original search to the final selection of studies is presented in Fig 1. In total, 17 studies (3875 participants) compared between extended resection and tumour resection alone [10, 19, 20, 22, 23, 31, 32, 38, 44, 45, 48, 51, 53, 56, 57, 59, 60]. Meanwhile, 26 studies (17368 patients) reported data on different surgical resection margins [12, 17, 18, 20, 21, 24–28, 33–36, 40–43, 46, 47, 49, 50, 52, 54, 55, 61], and five studies (1619 participants) compared the long-term outcomes between surgery and conservative treatment in patients with recurrent RPS [29, 30, 37, 39, 58]. The characteristics of the included studies are shown in Table 1. The overall risk of bias in this analysis was deemed low to moderate (S1 File).

### Extended resection versus tumour resection alone

A total of 17 studies reported data on extended resection versus tumour resection alone (Fig 2). Five studies compared the complications between extended resection and tumour resection alone [10, 23, 44, 51, 57]; however, one trial reported no events [44]. The overall complication rate was 21% (81/394). The pooled analysis of the four trials [10, 23, 51, 57] did not show a significant difference in complications between extended resection and tumour resection alone (44/184 vs. 33/210; OR: 2.24, 95% CI: 0.94–5.34; $P$ = 0.068; S1 File). Sensitive analyse that only including studies with similar surgical margins [10, 23, 51] showed that the extended resection group had a higher complication rate than the tumour resection alone group (OR: 3.61, 95% CI: 1.56–8.31; $P$ = 0.003).

Fatal outcomes related to operation were reported in seven trials (2643 participants) [10, 19, 22, 44, 51, 57, 59], but four of them reported no events in either group [10, 22, 44, 51]. Three studies [19, 57, 59] reported 44 surgery-related deaths (22 in the extended resection group and 22 in the tumour resection alone group). The overall surgery-related mortality rate was 2%. The pooled analysis of the three trials did not show a significant difference between the extended resection group and tumour resection alone group (OR: 1.00, 95% CI: 0.55–1.81; $P$ = 0.997; S1 File). Sensitive analyse that only including studies [59] with similar surgical margins also showed no significant difference between the extended resection group and tumour resection alone group (OR: 0.95, 95% CI: 0.57–1.76; $P$ = 0.877).

There were seven studies [20, 31, 32, 38, 45, 48, 53] (790 patients) that reported disease-free survival, and they were pooled in a random-effects model. The results showed no significant

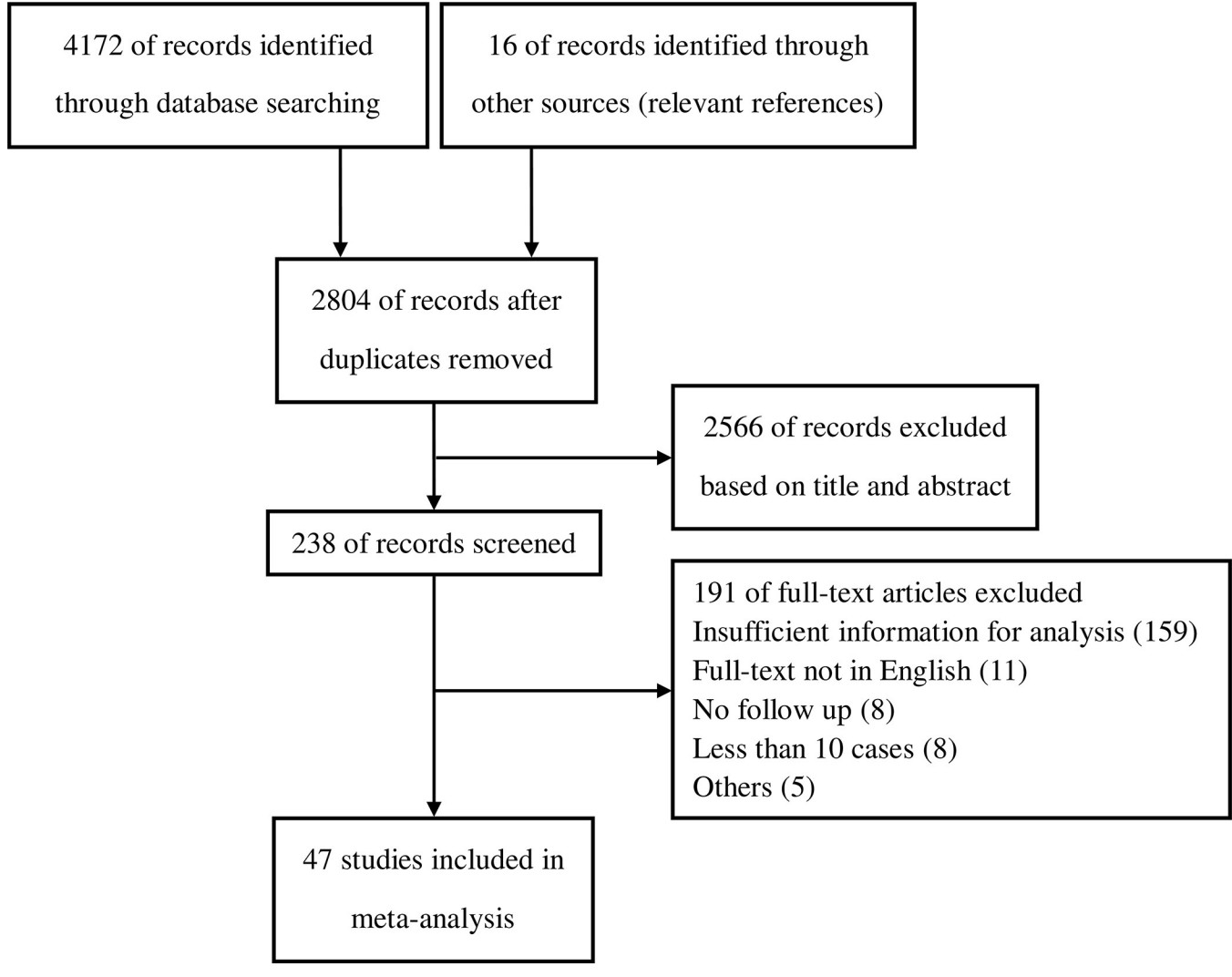

**Fig 1. PRISMA flow diagram.**

difference between the extended resection group and the tumour resection alone group (HR: 1.00; 95% CI: 0.74–1.27; $P$ = 0.945; S1 File), with low heterogeneity ($I^2$ = 23%). Subgroup analysis by tumour status also showed no significant differences in disease-free survival in the primary RPS (HR: 1.11; 95% CI: 0.68–1.53; $P$ = 0.645; $I^2$ = 14%; S1 File) or recurrent RPS (HR: 0.94; 95% CI: 0.45–1.44; $P$ = 0.676; $I^2$ = 68%; S1 File) subgroups. Sensitive analyse that only including studies with similar surgical margins [31, 38] also did not show significant difference between the extended resection group and tumour resection alone group (OR: 1.22, 95% CI: 0.84–1.60; $P$ = 0.409).

We pooled the results of 11 studies [19, 31, 32, 38, 44, 45, 51, 53, 56, 59, 60] (3014 patients) that reported HRs for OS. The results indicated no significant difference between the extended resection group and the tumour resection alone group (HR: 0.93; 95% CI: 0.83–1.03; $P$ = 0.774; S1 File), with no heterogeneity ($I^2$ = 0%). Subgroup analysis based on tumour status also did not show a significant difference between the two groups in primary RPS (HR: 0.94; 95% CI: 0.83–1.04; $P$ = 0.477; $I^2$ = 0%; S1 File) and in recurrent RPS (HR: 0.90; 95% CI: 0.00–1.81; $P$ = 0.531; $I^2$ = 0%; S1 File). Sensitive analyse that only including studies with similar surgical

**Table 1. Characteristics of the studies included for meta-analysis.**

| Study | Patients (n) | Histologic subtype (%) | FNCLCC grade (%) | RT (%) | CT (%) | Primary/ Recurrent (%) | Combined organ resection (%) | Margin status (%) | Vascular reconstruction (%) | Distant metastasis (%) |
|---|---|---|---|---|---|---|---|---|---|---|
| Abdelfatah 2016 [17] | 131 | Lip, 38; Lei, 40; MFH, 4 | G1, 18; G2, 21; G3, 53 | 24 | 28 | P, 100 | 82 | R0, 31; R1, 49; R2, 16 | 14 | 18 |
| Bagaria 2018 [18] | 5407 | Lip, 51; Lei, 23; MFH, 2 | NA | 26 | 17 | P, 100 | NA | R0, 69; R1, 26; R2, 5 | NA | NA |
| Bengmark 1990 [19] | 15 | Lip, 0; Lei, 33; MFH, 13 | NA | NA | NA | NA | 27 | NA | NA | NA |
| Bonvalot 2008 [20] | 382 | Lip, 50; Lei, 18; MFH, 9 | G1, 32; G2, 34; G3, 30 | NA | NA | P, 100 | 67 | R0, 47; R1, 26; R2, 10 | NA | 3 |
| Bremjit 2014 [21] | 132 | Lip, 61; Lei, 22 | G1, 38; G2, 34; G3, 27 | 30 | 21 | P, 100 | 76 | R0, 48; R1, 47; R2, 5 | 16 | NA |
| Chiappa 2006 [22] | 47 | Lip, 53; Lei, 28; MFH, 8 | NA | NA | NA | P, 49; R, 51 | 64 | R0, 60; R1, 6; R2, 34 | NA | NA |
| Chiappa 2018 [23] | 83 | Lip, 53; Lei, 28; MFH, 8 | NA | NA | NA | P, 55; R, 45 | 64 | R0, 74; R1, 19; R2, 7 | NA | NA |
| Doepker 2016 [24] | 35 | Lip, 26; Lei, 26 | G1, 34; G2, 6; G3, 60 | 38 | 23 | P, 100 | NA | R0, 49; R1, 28; R2, 3 | NA | NA |
| Erzen 2005 [25] | 102 | Lip, 28; Lei, 37; MFH, 7 | G1, 40; G2, 18; G3, 41 | NA | NA | P, 55; R, 45 | NA | R0, 54; R1, 41; R2, 3 | 12 | NA |
| Fujimoto 2018 [26] | 167 | Lip, 33; Lei, 6 | NA | NA | 4 | P, 100 | 41 | R0/R1, 89; R2, 11 | NA | NA |
| Garcı́a-Aceituno 2010 [27] | 46 | Lip, 35; Lei, 11; MFH, 11 | G1, 59; G2, 13; G3, 28 | 17 | 2 | P, 100 | 30 | R0, 59; R1, 19; R2, 22 | NA | NA |
| Gilbeau 2002 [28] | 93 | Lip, 58; Lei, 18; MFH, 16 | G1, 29; G2, 47; G3, 24 | 100 | 24 | P, 100 | NA | R0, 38; R1, 58; R2, 4 | NA | NA |
| Grobmyer 2010 [29] | 78 | Lip, 54; Lei,19 | G1, 47; G2, 13; G3, 36 | 66 | 13 | R, 100 | 39 | R0/R1, 60; R2, 16 | NA | 21 |
| Gronchi 2014 [30] | 377 | Lip, 63; Lei, 16; MFH, 4 | G1, 36; G2, 36; G3, 28 | 32 | 31 | P, 100 | 93 | NA | NA | NA |
| Ikoma 2017 [31] | 172 | Lip, 100 | G1, 5; G2, 17; G3, 48 | 20 | 40 | P, 100 | 70 | R0, 65 | 21 | NA |
| Ikoma 2018 [10] | 83 | Lip, 100 | NA | NA | NA | P, 100 | 46 | R0/R1, 92; R2, 8 | NA | NA |
| Ishii 2020 [32] | 52 | Lip, 100 | NA | NA | NA | P, 100 | 78 | R0, 35 | NA | 6 |
| Jaques 1989 [33] | 146 | Lip, 50; Lei, 29; MFH, 4 | NA | NA | NA | P, 55; R, 45 | 83 | R0/R1, 59; R2, 15 | NA | NA |
| Karakousis 1985 [34] | 68 | Lip, 32; Lei, 32 | NA | NA | NA | P, 100 | NA | R0,/R1, 40; R2, 10 | NA | NA |
| Lehnert 2009 [35] | 110 | Lip, 54; Lei, 23 | G1, 22; G2, 26; G3, 53 | NA | NA | P, 65; R, 35 | 58 | R0, 35; R1, 33; R2, 23 | NA | NA |
| Lewis 1998 [36] | 500 | Lip, 41; Lei, 27; MFH, 7 | NA | NA | NA | P, 56; R, 44 | | R0, 42; R1, 17; R2, 18 | NA | 20 |
| Lochan 2011 [37] | 75 | Lip, 32 | G1, 60; G2, 40 | NA | NA | P, 96; R, 4 | NA | R0, 68; R1, 32 | NA | NA |
| Lu 2013 [38] | 19 | Lip, 100 | NA | NA | NA | R, 100 | 21 | R0, 79; R1, 16; R2, 5 | NA | NA |
| MacNeill 2017 [39] | 408 | Lip, 63; Lei, 25 | G1, 16; G2, 40; G3, 42 | 15 | 43 | R, 100 | NA | NA | NA | 46 |
| Martin 2020 [40] | 43 | NA | NA | 21 | 19 | P, 100 | NA | R0, 28; R1, 21; R2, 5 | NA | NA |
| McGrath 1984 [41] | 47 | Lip, 28; Lei, 32; MFH, 17 | NA | NA | NA | P, 100 | NA | R0/R1, 38; R2, 62 | NA | NA |

(*Continued*)

**Table 1.** (Continued)

| Study | Patients (n) | Histologic subtype (%) | FNCLCC grade (%) | RT (%) | CT (%) | Primary/ Recurrent (%) | Combined organ resection (%) | Margin status (%) | Vascular reconstruction (%) | Distant metastasis (%) |
|---|---|---|---|---|---|---|---|---|---|---|
| Milone 2011 [42] | 32 | Lip, 100 | NA | NA | NA | NA | NA | R0, 66; R1, 19 | NA | NA |
| Miura 2015 [43] | 8653 | Lip, 46; Lei, 24 | G1, 27; G2, 12; G3, 23 | 26 | 18 | NA | NA | R0, 48; R1, 15; R2, 15 | NA | NA |
| Morizawa 2006 [44] | 23 | Lip, 52; Lei, 17; MFH, 14 | G1, 14; G2, 17; G3, 69 | NA | NA | P, 100 | 61 | R0, 17; R1, 74; R2, 9 | NA | NA |
| Mussi 2011 [45] | 77 | Lip, 39; Lei, 26 | G1, 33; G2, 27; G3, 40 | 30 | 35 | P, 100 | 65 | R0/R1, 88 | NA | NA |
| Nathenson 2018 [46] | 49 | Lip, 57; Lei, 43 | G1, 33; G2, 14; G3, 49 | 37 | NA | P, 41; R, 59 | NA | R0, 47; R1, 31; R2, 6 | NA | NA |
| Pinson 1989 [47] | 79 | Lip, 27; Lei, 13; MFH, 9 | NA | NA | NA | P, 100 | NA | R0/R1, 48; R2, 20 | NA | NA |
| Rhu 2019 [48] | 74 | Lip, 100 | G1, 36; G2, 40; G3, 24 | 42 | NA | R, 100 | 70 | NA | NA | NA |
| Roeder 2017 [49] | 156 | Lip, 61; Lei, 17 | G1, 11; G2, 33; G3, 56 | NA | NA | P, 44; R, 56 | NA | R0, 27; R1, 65; R2, 8 | NA | NA |
| Rossi 2013 [50] | 78 | Lip, 55; Lei, 22 | G1, 44; G2, 20; G3, 36 | NA | NA | P55; R45 | NA | R0, 19; R1, 74; R2, 6 | NA | NA |
| Santos 2010 [51] | 91 | Lip, 31; Lei, 32 | G1/G2, 40; G3, 60 | NA | NA | NA | 60 | R0, 46; R1/R2, 54 | NA | NA |
| Shibata 2001 [12] | 55 | Lip, 100 | NA | NA | NA | P, 53; R, 47 | | R2, 78 | NA | NA |
| Shiloni 1993 [52] | 41 | Lip, 24; Lei, 24; MFH, 15 | NA | 41 | 71 | P, 51; R, 49 | 51 | R0/R1, 54; R2, 37 | NA | 17 |
| Singer 2003 [53] | 177 | Lip, 100 | NA | 8 | 0 | P, 100 | 26 | R0, 44; R1, 37; R2, 19 | NA | NA |
| Tan 2016 [54] | 675 | Lip, 50; Lei, 23; | NA | 8 | 18 | P, 100 | 58 | R0, 50; R1, 35; R2, 9 | 10 | NA |
| Thalji 2020 [55] | 70 | Lip, 24; Lei, 19 | G1, 11; G2, 74; G3, 15 | 10 | 51 | P, 31; R, 69 | NA | R0, 23; R1, 15; R2, 58 | NA | NA |
| Tropea 2020 [56] | 51 | Lip, 62; Lei, 18 | G1, 26; G2, 10; G3, 64 | 78 | 45 | R, 100 | 59 | R0, 37; R1, 59; R2, 4 | NA | NA |
| Tseng 2010 [57] | 156 | NA | NA | 12 | 1 | NA | 37 | NA | 4 | NA |
| van Houdt 2020 [58] | 681 | Lip, 80; Lei, 8 | G1, 28; G2, 26; G3, 40 | 13 | 36 | R, 100 | NA | R0/R1, 83; R2, 15 | NA | 19 |
| Villano 2020 [59] | 2278 | Lip, 54; Lei, 25 | G1, 42; G2, 19; G3, 39 | NA | NA | P, 100 | 50 | R0/R1, 87 R2, 2 | NA | NA |
| Yang 2015 [60] | 95 | Lip, 47; Lei, 27 | G1, 28; G2, 31; G3, 32 | 35 | 42 | NA | 55 | R0/R1, 87 | NA | NA |
| Zhao 2015 [61] | 71 | Lip, 100 | NA | NA | NA | P, 100 | 31 | R0, 55; R1, 31; R2, 14 | NA | NA |

Abbreviations: CT, Chemotherapy; Lip, Liposarcoma; Lei, Leiomyosarcoma; MFH, Malignant fibrous histiocytoma; P, Primary; R, Recurrent; RT, Radiotherapy; NA, data not available; R status, Resection status.

margins [31, 38, 44, 51, 59] also showed no significant difference between the extended resection group and tumour resection alone group (OR: 0.93, 95% CI: 0.83–1.04; $P$ = 0.951).

## Surgical resection margins

There were 26 studies that reported data on outcomes by different surgical resection margins (Fig 3). In 17 studies [18, 20, 21, 24, 25, 27, 28, 35, 40, 42, 43, 46, 49, 50, 54, 55, 61] (16357

Extended resection vs tumour resection alone

| Study | No. of studies | No. of patients | | HR/OR (95% CI) | I-squared |
|---|---|---|---|---|---|
| | | | Favors tumour resection alone / Favors extended resection | | |
| Complication rate | 5 | 394 | | 2.24 (0.94, 5.34) | 50.5% |
| Mortality rate | 7 | 2643 | | 1.00 (0.55, 1.81) | 0.0% |
| OS | 11 | 3014 | | 0.93 (0.83, 1.03) | 0.0% |
| OS (primary) | 6 | 2779 | | 0.94 (0.83, 1.04) | 0.0% |
| OS (recurrent) | 2 | 87 | | 0.90 (0.40, 1.81) | 0.0% |
| DFS | 7 | 790 | | 1.00 (0.74, 1.27) | 23.1% |
| DFS (primary) | 5 | 684 | | 1.11 (0.68, 1.53) | 14.4% |
| DFS (recurrent) | 2 | 106 | | 0.94 (0.45, 1.44) | 67.8% |

0.1  0.2  0.5  1  2  5  10

**Fig 2. Meta-analysis results of extended resection versus tumour resection alone.**

patients), there was a significant difference in OS between R0 and R1, with a pooled HR of 1.34 (95% CI: 1.23–1.44; $P < 0.001$; $I^2 = 0\%$; S1 File). In subgroup analysis by tumour status, the pooled analysis of nine trials [18, 20, 21, 24, 27, 28, 40, 54, 61] on primary RPS showed that R1 resection has an inferior OS rate to R0 resection (HR: 1.31; 95% CI: 1.19–1.43; $P < 0.001$; S1 File), with no heterogeneity ($I^2 = 0\%$). Meanwhile, 7 studies [21, 27, 35, 46, 49, 54, 61] (1239 patients) compared the OS between R1 and R2 resection. The results showed that R1 resection achieves superior OS (HR: 1.86; 95% CI: 1.35–2.36; $P < 0.001$; $I^2 = 10\%$; S1 File). The benefit of was also significant in the subgroup of trials reporting primary RPS (HR: 1.77; 95% CI: 1.05–2.50; $P = 0.01$; $I^2 = 36\%$; S1 File).

A total of 12 studies [12, 17, 26, 27, 33–36, 41, 46, 47, 52] (1510 patients) compared survival outcomes between R2 resection and no surgery. The results showed that R2 resection achieves superior OS to no surgery (HR: 1.26; 95% CI: 1.07–1.45; $P < 0.001$; $I^2 = 7\%$; S1 File). However, in the studies on primary RPS, the pooled HR was similar between the R2 resection and no surgery groups (HR: 1.14; 95% CI: 0.87–1.42; $P = 0.422$; S1 File).

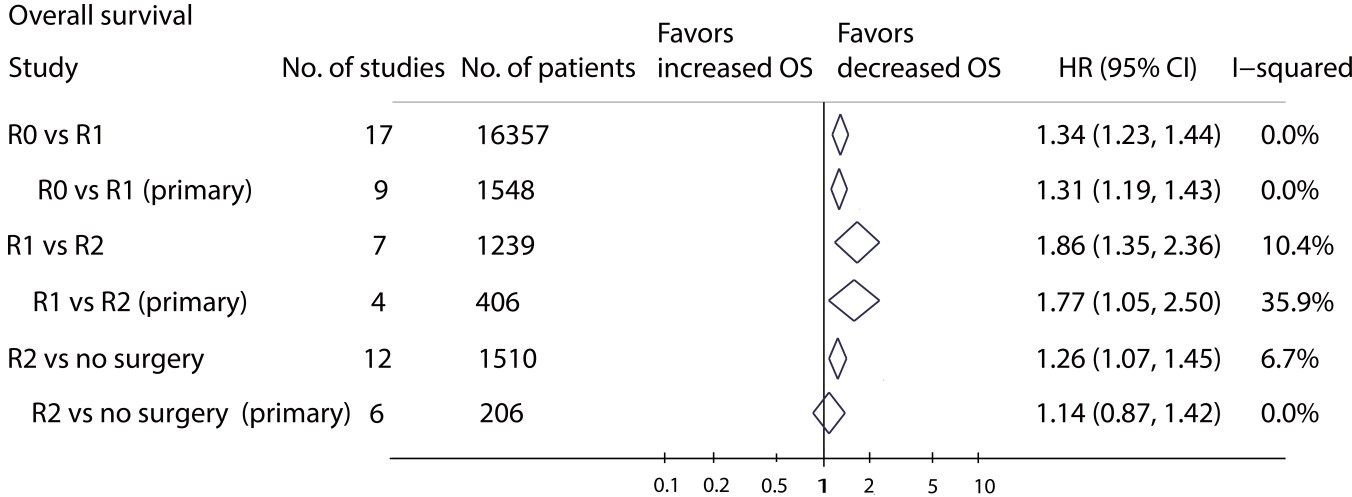

Overall survival

| Study | No. of studies | No. of patients | | HR (95% CI) | I-squared |
|---|---|---|---|---|---|
| | | | Favors increased OS / Favors decreased OS | | |
| R0 vs R1 | 17 | 16357 | | 1.34 (1.23, 1.44) | 0.0% |
| R0 vs R1 (primary) | 9 | 1548 | | 1.31 (1.19, 1.43) | 0.0% |
| R1 vs R2 | 7 | 1239 | | 1.86 (1.35, 2.36) | 10.4% |
| R1 vs R2 (primary) | 4 | 406 | | 1.77 (1.05, 2.50) | 35.9% |
| R2 vs no surgery | 12 | 1510 | | 1.26 (1.07, 1.45) | 6.7% |
| R2 vs no surgery (primary) | 6 | 206 | | 1.14 (0.87, 1.42) | 0.0% |

0.1  0.2  0.5  1  2  5  10

**Fig 3. Meta-analysis results of different surgical resection margins.**

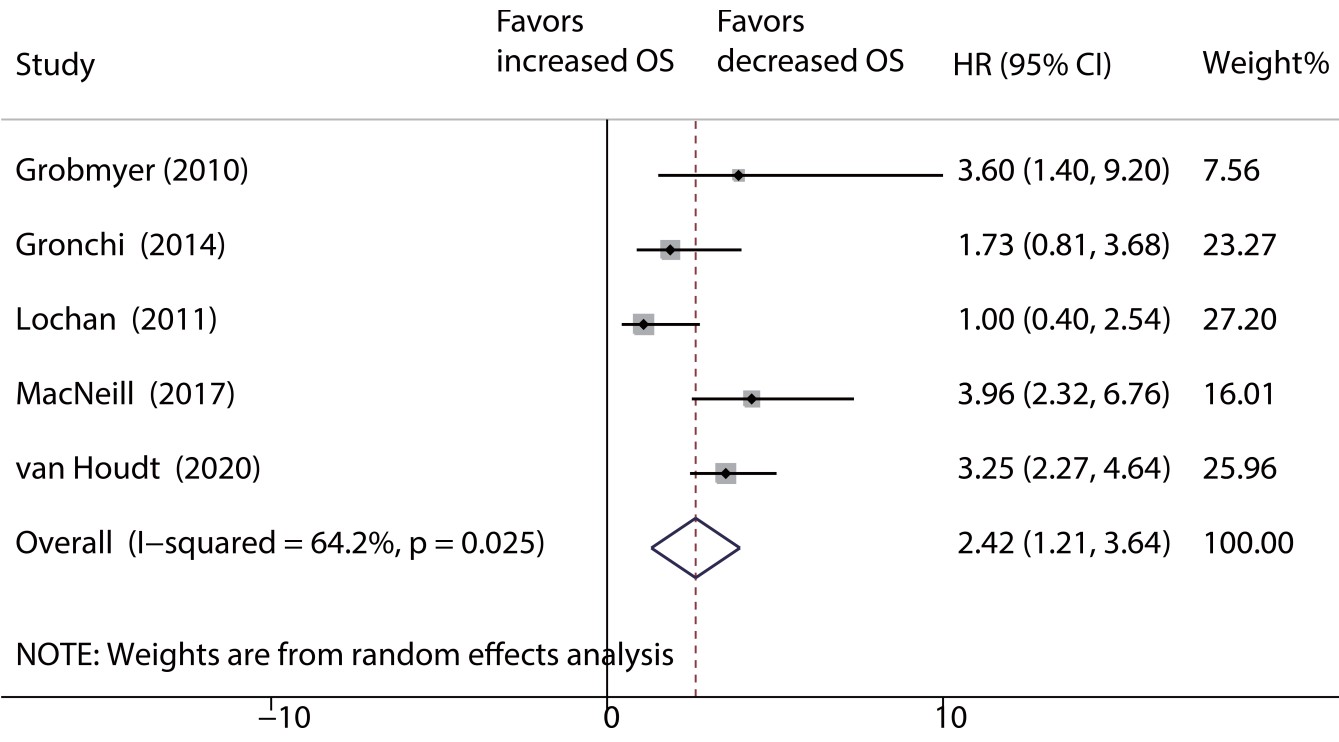

**Fig 4. Pooled over-all survival of surgery versus conservative treatment for recurrent RPS.**

### Impact of surgery on long-term survival in recurrent RPS

A total of five studies [8, 29, 37, 39, 58] reported data on surgery versus conservative treatment for recurrent RPS. The results from these studies demonstrated that surgical treatment achieves a significantly higher OS rate than does conservative treatment (HR: 2.42; 95% CI: 1.21–3.64; $P < 0.001$; Fig 4), with moderate to high heterogeneity ($I^2 = 64\%$).

## Discussion

Guidelines on the management of RPS are still lacking owing to its low incidence. In this study, an aggressive surgical approach with contiguous organ resection achieved acceptable rates of postoperative complication and mortality in both primary and recurrent RPS. The results of this study also demonstrated the importance of surgery and surgical margins in long-term survival. To our best knowledge, this is the largest and most comprehensive meta-analysis focusing on the role of surgery in RPS.

The first consensus on the management of primary RPS was published by the trans-Atlantic RPS working group (TARPSWG) in 2015 [62]. In the follow-up, the group included several more European and North American centres and further improved the consensus on recurrent and metastatic RPS [63, 64]. However, definitive evidence to guide clinical practice is still lacking. Multimodality treatment involving radiotherapy and/or chemotherapy is recommended to obtain negative surgical margins with a subsequently better local disease control and longer survival in STS in the extremity [4]. However, the use of adjuvant radiotherapy and chemotherapy in STS in the retroperitoneum varies widely among institutions because of the lack of high-level evidence supporting the benefit of these modalities [62, 65]. A meta-analysis of ten non-RCTs concluded that perioperative radiation therapy is associated with higher OS and lower recurrence rates [66]. However, a recent multicentre RCT that compared between

preoperative radiotherapy plus surgery and surgery alone for patients with primary RPS reported conflicting results [7]. There are also limited evidence on the usefulness of neoadjuvant therapy for recurrent RPS patients indicated for resection. In addition, radiotherapy to the retroperitoneum is a complex procedure. RCTs are needed to standardise the radiotherapy protocol for recurrent/unresectable RPS.

Given the lack of data, surgical resection remains the cornerstone of therapy and the only potentially curative therapy for patients with RPS. However, many aspects of surgical resection for RPS are controversial. For example, the efficacy of contiguous organ resection and the appropriate extent of curative-intent surgical resection are yet to be determined. Further, the role of gross incomplete resection for unresectable RPS needs to be clarified. The criteria for unresectability remains undefined, and the indication and eligibility for surgical resection vary by medical centre. Patients with residual macroscopic disease are often referred to specialised centres because they are a significant challenge from a surgical standpoint as the appropriateness of en bloc resection for organs adherent to the tumour needs to be determined intraoperatively [67]. The TARPSWG recently updated the consensus on management of primary RPS in adults [68]. The update mentioned criteria for technical non-resectability as involvement of the superior mesenteric artery, aorta, coeliac trunk, and/or portal vein; bone involvement; growth into the spinal canal; invasive extension of retrohepatic inferior vena cava leiomyosarcoma into the right atrium; infiltration of multiple major organs (eg, liver and pancreas) and/or major vessels. However, vascular reconstructions, which enable radical resection of retroperitoneal sarcomas in patients with advanced disease, have been successfully performed in many studies [69, 70]. Further, complex surgeries are associated with an acceptable rate of serious perioperative complications [69]. In addition, a previous study indicated that more than one third of the patients with primary/recurrent RPS undergoing palliative-intent operation could achieve R0/R1 resection [31]. Thus, unresectability cannot be determined via computed tomography imaging alone, and patients should be referred to specialised centres and carefully evaluated by an experienced multidisciplinary team before any surgical resection is attempted. Furthermore, our results showed that even R2 resection achieves superior OS to no surgery, and surgical treatment achieves a significantly higher OS rate than does conservative treatment in recurrent RPS. These findings indicate that surgical resection should be considered as first-line treatment regardless of the tumour status (primary or recurrent).

With respect to the impact of organ resection, our findings indicated that rates of postoperative mortality are not significantly different between extended resection group and tumour resection alone, however, extended resection group had a relatively higher complication rate than the tumour resection alone group. In addition, organ resection did not improve local recurrence or OS. Given the importance of a quality surgical resection, early techniques ascribed to an aggressive surgical approach whereby adjacent uninvolved organs are routinely resected en bloc to optimise the margin status [20]. These techniques are referred to as compartmental resection [20]. Complete compartmental resection is defined as a systematic resection of uninvolved contiguous organs [20]. In general, the patient undergoes an en bloc tumour resection with the colon in front, the kidney inside, and the psoas at the back. Vessels are exposed after removal of adventitia, but the pancreas and duodenum are not resected if they are not involved. In contrast, contiguous organ resection is defined as resection of macroscopically involved adjacent organs [20]. Theoretically, complete compartmental resection could obtain a rim of normal tissue surrounding the tumour to ensure a better margin. However, compartmental resection only results in a lower local recurrence rate and is associated with a higher overall complication and lesser survival benefit than complete resection and contiguous organ resection [20, 51]. These results might be explained by the following reasons. Both compartmental resection and contiguous organ resection have no impact on surgical

resection margins, especially R0 resection [51]. The R0 resection is only approximately 57% in compartmental resection [51]. Unlike the more common epithelial tumours or adenocarcinomas, which develop within a single organ, RPS can infiltrate multiple surrounding organs owing to their large size and multiple central location [9, 44]. Tumours measuring 20 cm on average have poorly defined anatomic borders, and thus, it would impractical to assess margin status [4]. In addition, it is challenging to obtain clear margins because RPS tumours are commonly surrounded by both anterior and posterior great vessels, vertebral column, and lumbar musculature. As such, although complete macroscopic surgical resection can be achieved in RPS, the incidence of local recurrence and disease progression remains high [39]. Determining the need for resection of adjacent organs depends on the surgeon's assessment of the extent of tumour invasion. Thus, understanding the survival benefit of radical excision of adjacent organs is crucial. As such, it is important that the need for extended resection is recognised pre/intraoperatively by multidisciplinary evaluation.

Consistent with previous studies [35, 46], we found that surgical resection margins are correlated with long-term survival. The current meta-analysis indicated that OS was higher in R0 resection than in R1 resection and in R1 resection than in R2 resection. Similar findings were obtained in subgroup analysis by tumour status. R2 resection achieved a superior OS to no surgery. However, interestingly, the pooled HR in the studies on primary RPS showed a similar OS between the R2 resection group and the no operation group. This could be because patients with unresectable primary RPS might have higher TNM stage or histological grade, which could be associated with worse long-term outcomes. Thus, for these patients, owing to the similar rates of postoperative complication and mortality between extended resection and tumour resection alone, adjacent organs with evidence of direct invasion must be resected en bloc to avoid R2 resection. In contrast, incomplete surgical resection was beneficial for patients with recurrent RPS, prolonging survival and alleviating symptoms [12].

The strengths of our review include its comprehensive search and methodologic robustness. We searched all available literature to exclude studies with overlapping cohorts and analysed large-scale studies. However, the present study also had some limitations. First, selection bias is inevitably associated with this type of surgical studies, especially when the indication and eligibility for surgical resection and the method of assessment of appropriate resection margins might vary by medical centre. The FNCLCC grade, tumour status, and adjuvant therapy also varied among the studies, possibly introducing bias. Although we performed subgroup analysis to investigate the impact of tumour status, we were unable to evaluate other factors that may modify the association between different surgical strategies and survival outcomes (eg, histologic subtype and adjuvant therapy) because the relevant data were lacking. Second, there was an insufficient number of studies on extended resection (eg, adjacent organs vs tumour resection alone) and surgical treatment vs conservative treatment for recurrent RPS were insufficient, and thus, the recommendations for these comparisons have a relatively weak power. Subsequent long-term prospective studies in these areas are needed. Third, the included studies were limited to the literatures published in English. This strategy might lead to limited data collection. Finally, all trials included in this study used an open-label design, which might introduce bias. However, assessment of the methodological quality of the included studies indicated that most studies had a low or medium risk of bias.

In summary, RPS is a rare and complex malignancy that is best managed by an experienced multidisciplinary team in a specialised referral centre. Surgical resection should be attempted in majority of the patients. Primary RPS should be indicated for curative-intent en bloc resection with optimal extent of resection, and adjacent organs with evidence of direct invasion must be resected en bloc to avoid R2 resection. Routine compartmental resection is not

recommended. Meanwhile, a part of unresectable recurrent RPS should be indicated for incomplete resection or debulking to improve survival after multidisciplinary evaluation.

## Supporting information

**S1 File. It contains all the supporting tables and figures.**
(DOC)

## Acknowledgments

The authors thank D.Y. Kang, statistician of the Department of Evidence-based Medicine and Clinical Epidemiology, West China Hospital, Sichuan University, Chengdu, for his assistance with the statistical analysis.

## Author Contributions

**Conceptualization:** Qiang Guo, Bin Huang.

**Data curation:** Qiang Guo, Jichun Zhao, Xiaojiong Du.

**Formal analysis:** Qiang Guo, Jichun Zhao, Xiaojiong Du.

**Investigation:** Qiang Guo, Jichun Zhao, Xiaojiong Du, Bin Huang.

**Methodology:** Qiang Guo, Jichun Zhao, Xiaojiong Du, Bin Huang.

**Project administration:** Qiang Guo, Jichun Zhao, Xiaojiong Du.

**Resources:** Qiang Guo, Jichun Zhao, Xiaojiong Du, Bin Huang.

**Software:** Qiang Guo.

**Supervision:** Xiaojiong Du, Bin Huang.

**Validation:** Qiang Guo.

**Visualization:** Qiang Guo.

**Writing – original draft:** Qiang Guo.

**Writing – review & editing:** Qiang Guo, Jichun Zhao, Xiaojiong Du, Bin Huang.

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
