## [Decision Letter · Decision Letter 0]

6 May 2022

PONE-D-21-35277

Survival Outcomes of Surgery for Retroperitoneal Sarcomas: A Systematic Review and Meta-analysis

PLOS ONE

Dear Dr. Bin Huang,

Thank you for submitting your manuscript to PLOS ONE. After careful consideration, we feel that it has merit but does not fully meet PLOS ONE’s publication criteria as it currently stands. Therefore, we invite you to submit a revised version of the manuscript that addresses the points raised during the review process.

We look forward to receiving your revised manuscript.

Kind regards,

Paolo Aurello

Academic Editor

PLOS ONE

2. PLOS requires an ORCID iD for the corresponding author in Editorial Manager on papers submitted after December 6th, 2016. Please ensure that you have an ORCID iD and that it is validated in Editorial Manager. To do this, go to ‘Update my Information’ (in the upper left-hand corner of the main menu), and click on the Fetch/Validate link next to the ORCID field. This will take you to the ORCID site and allow you to create a new iD or authenticate a pre-existing iD in Editorial Manager. Please see the following video for instructions on linking an ORCID iD to your Editorial Manager account: https://www.youtube.com/watch?v=_xcclfuvtxQ.

3. We note that this manuscript is a systematic review or meta-analysis; our author guidelines therefore require that you use PRISMA guidance to help improve reporting quality of this type of study. Please upload copies of the completed PRISMA checklist as Supporting Information with a file name “PRISMA checklist”.

Reviewers' comments:

Reviewer's Responses to Questions

**Comments to the Author**

1. Is the manuscript technically sound, and do the data support the conclusions?

Reviewer #1: Partly

Reviewer #2: Partly

Reviewer #3: Partly

2. Has the statistical analysis been performed appropriately and rigorously? 

Reviewer #1: Yes

Reviewer #2: N/A

Reviewer #3: I Don't Know

3. Have the authors made all data underlying the findings in their manuscript fully available?

Reviewer #1: Yes

Reviewer #2: Yes

Reviewer #3: Yes

4. Is the manuscript presented in an intelligible fashion and written in standard English?

Reviewer #1: Yes

Reviewer #2: Yes

Reviewer #3: Yes

5. Review Comments to the Author

Reviewer #1: Survival Outcomes of Surgery for Retroperitoneal Sarcomas: A Systematic Review and Meta-analysis

First of all, I would like to congratulate the authors for this systematic review and meta-analysis. However, I would like them to clarify following:

1. In the Study characteristics section, they describe: „After removing the 1384 duplicates, the titles and abstracts of 2804 articles were reviewed. Among them, 238 studies were reviewed in full text. Finally, 47 studies involving 22608 patients were included in the final analysis“ What was the reaseon for excluding these 2757 records?

2. Could the author include data regarding the tumor location: for example right or left hemiabdomen ?

3. I suggested that the included studies were limited to the literatures published in English. Please include that to the limitations.

4. In the discussion part, the authors puts forward the limitations oft he research. Can the authors describe how to overcome these limitations?

5. In summery section:“ Meanwhile, unresectable recurrent RPS should be indicated for incomplete resection or debulking to improve survival.“ No data in the meta-analysis determined this point: unresectable recurrent RPS vs no surgery. Please indicate.

Reviewer #2: I want to congratulate the authors for such meticulous and important work.

As introduction: STS are more than 70 different subtypes I believe that´s important to remark that Retroperitoneal soft tissue sarcomas (RPS) are rare tumors that include several well-defined histologic subtypes. The most common histology is liposarcoma, followed by leiomyosarcoma. Other rare subtypes include MFH, solitary fibrous tumors and malignant peripheral nerve sheath tumors.

About the Impact of surgery on long-term survival in recurrent RPS.

Patients from the study of Gronchi are also reported in the study of MacNeill 2017 (that includes patients from IRCCS Foundation National Cancer Institute (Milan, Italy) and Gustave Roussy Institute (Villejuif, France).

In the discussion of this last study, they stated in the discussion that “An earlier collaboration between 2 of the participating centers in the current series suggested that after extended resection for primary disease, the benefit of reoperation for LR may be limited.”

On the other hand, the study of Houdt only includes patients with a second recurrence.

About the lack of guidelines, the Consensus on the management of primary RPS published in 2015 by the Trans-Atlantic RPS Working Group (TARPSWG), updated for the last time in 2021 , I think are an important guide in decision making process.

The use of morbidity in the studies reviewed are more related with surgery complications than with long-term morbidity. A nice article that deals with this subject was published, I share the reference:

Severe chronic pain and lower limb motor impairment after multivisceral resection for retroperitoneal sarcomas are rare. Long-term renal function is not significantly impaired when nephrectomy is performed -- Callegaro D, Miceli R, Brunelli C, Colombo C, Sanfilippo R, Radaelli S, Casali PG, Caraceni A, Gronchi A, Fiore M. Long-term morbidity after multivisceral resection for retroperitoneal sarcoma. Br J Surg. 2015 Aug;102(9):1079-87. doi: 10.1002/bjs.9829. Epub 2015 Jun 3. PMID: 26041724.

About the use of adjuvant radiotherapy in STS in the retroperitoneum the lack of high-level evidence supporting the benefit of these modality is crucial, but there are issues regarding acute and long-term toxicity not making this and absolute indication even when R2 resections are performed.

I agree that the indication and eligibility for surgical resection vary by medical center, but as you mentioned TARPSWG criteria give some backbones for decision making process, and as you say all patients must be referred to specialised centres because they are a significant challenge from a surgical standpoint.

When you mark the bias of this study I want to add

- Selection bias that is inevitably associated with this type of surgical studies

- Complexity to define R0 vs R1 in different retrospective and multi-institutional studies

- Do not take in account histological subtype and there different behaviours.

For example, well-differentiated liposarcoma essentially displayed a local risk of 20-40 % at 5 years after extended primary resection, and in some important series no patient developed metastatic disease.

On the other hand, leiomyosarcoma displayed a predominant systemic risk (greater than 40-50 %), with an extended primary approach optimized local control (95 % at 5 years in some series), but it could not prevent distant spread. Leiomyosarcoma had a better post-distant metastases outcome compared to other histologic subtypes.

About your finals conclusions, I agree with them but not with the last one.

Meanwhile, unresectable recurrent RPS should be indicated for incomplete resection or debulking to improve survival -- I think that the indication for surgery after recurrence is very complex decision, and should be discussed on a multidisciplinary team individual cases considering histologic subtype and grade, disease free interval, presence of distant metastases, tumor multifocality, limited performance status, and complex multivisceral resections.

In the study of Grobney 48/61 with LR only underwent surgery.

Grochi wrote in the discussion that "Unlike the primary resection, the second surgery was not intended to remove adherent organs if not directly infiltrated, because there is no chance of cure after recurrence."

In the study of Lochan et al, 22/46 LR underwent surgery and there is no data about histologic type or grade.

ManNeill reported initial site of recurrence was local only (LR) for 219 patients, of whom 105 patients (48%) underwent surgery.

Reviewer #3: The background to the study was well laid and rationale. The presentation and structure of the abstract was however below par and could be improved. The manuscript was presented in standard English and made-for-easy reading. The intellectual content though satisfactory could be improved as suggested in the attached file.

6. PLOS authors have the option to publish the peer review history of their article (what does this mean?). If published, this will include your full peer review and any attached files.

Reviewer #1: No

Reviewer #2: **Yes: **Eduardo Daniel Adelchanow

Reviewer #3: No

---

## [Author Response · Author response to Decision Letter 0]

13 Jun 2022

PONE-D-21-35277

Review Comments to the Author

Reply to reviewer #1

Dear reviewer:

We appreciate your attitude to scientific review process and thank you for your comments. 

Reviewer #1: Survival Outcomes of Surgery for Retroperitoneal Sarcomas: A Systematic Review and Meta-analysis

First of all, I would like to congratulate the authors for this systematic review and meta-analysis. However, I would like them to clarify following:

1. In the Study characteristics section, they describe: „After removing the 1384 duplicates, the titles and abstracts of 2804 articles were reviewed. Among them, 238 studies were reviewed in full text. Finally, 47 studies involving 22608 patients were included in the final analysis “What was the reason for excluding these 2757 records?

Response: The inclusion and exclusion process for identified articles is shown in Fig. 1. According to the inclusion and exclusion criteria mentioned in the methods, after reviewing the title and abstract of the records, 2566 records were excluded because they were irrelevant articles. Besides, 159 articles were excluded because they can’t provide sufficient data for final analysis. The other 32 records were excluded because they met exclusion criteria (article not in English, no follow up, less than 10 cases, etc.).

2. Could the author include data regarding the tumor location: for example right or left hemiabdomen ?

Response: Of the included 47 studies, only one study mentioned the data regarding the tumor location. Gilbeau et al. (2002) reported a series of 45 patients, and they were equal in the location. Thus, we can’t include sufficient data regarding the tumor location.

3. I suggested that the included studies were limited to the literatures published in English. Please include that to the limitations.

Response: We have included that to the revised paper. 

4. In the discussion part, the authors puts forward the limitations of the research. Can the authors describe how to overcome these limitations?

Response: Since retroperitoneal soft tissue sarcomas (RPS) are rare tumors, reports on this topic are limited. As we described in the discussion, we tried to perform subgroup analysis to investigate the impact of different histologic subtype, tumour status, and adjuvant therapy, however, the relevant data were lacking. Besides, there was an insufficient number of studies on some comparisons. In all, it seemed difficult to overcome the limitations of this research.

5. In summery section: “Meanwhile, unresectable recurrent RPS should be indicated for incomplete resection or debulking to improve survival. “No data in the meta-analysis determined this point: unresectable recurrent RPS vs no surgery. Please indicate.

Response: In the last paragraph of the part of Results, we described that “A total of five studies reported data on surgery versus conservative treatment for recurrent RPS. The results from these studies demonstrated that surgical treatment achieves a significantly higher OS rate than does conservative treatment (HR: 2.42; 95% CI: 1.21–3.64; P < 0.001; Fig 4), with moderate to high heterogeneity (I2 = 64%).” These results might indicate this conclusion.

 

Reply to reviewer #2

Dear reviewer:

We appreciate your attitude to scientific review process and thank you for your comments. 

Reviewer #2: I want to congratulate the authors for such meticulous and important work.

As introduction: STS are more than 70 different subtypes I believe that´s important to remark that Retroperitoneal soft tissue sarcomas (RPS) are rare tumors that include several well-defined histologic subtypes. The most common histology is liposarcoma, followed by leiomyosarcoma. Other rare subtypes include MFH, solitary fibrous tumors and malignant peripheral nerve sheath tumors.

Response: We have added the above information in the revised paper.

About the Impact of surgery on long-term survival in recurrent RPS.

Patients from the study of Gronchi are also reported in the study of MacNeill 2017 (that includes patients from IRCCS Foundation National Cancer Institute (Milan, Italy) and Gustave Roussy Institute (Villejuif, France).

In the discussion of this last study, they stated in the discussion that “An earlier collaboration between 2 of the participating centers in the current series suggested that after extended resection for primary disease, the benefit of reoperation for LR may be limited.”

On the other hand, the study of Houdt only includes patients with a second recurrence.

Response: The study of Gronchi (2016) and the study of MacNeill (2017) shared the same study population (1007 cases). However，they analyzed different aspects of recurrent RPS. In the discussion of this last study, they stated in the discussion that “An earlier collaboration between 2 of the participating centers in the current series suggested that after extended resection for primary disease, the benefit of reoperation for LR may be limited.” This mentioned study was the study of Gronchi (2014), which consisted of 377 patients. They also complimented that “However, this was based on a small number of patients. In the current study of a much larger cohort, we demonstrate a significant association between resection and survival for patients with locally recurrent disease.”, which was consistent with our conclusion.

The study of Houdt (2020) also shared an initial series of 1007 study population. However, the follow-up period of this study was much longer than the study of MacNeill (2017). In the study of Houdt (2020), second recurrences occurred in 400 of 567 patients after an R0/R1 resection of a first locally recurrent RPS. In the study of MacNeill (2017), the population was 408 patients developed first recurrent disease during the follow-up period. Thus, the study population were not duplicate. 

About the lack of guidelines, the Consensus on the management of primary RPS published in 2015 by the Trans-Atlantic RPS Working Group (TARPSWG), updated for the last time in 2021 , I think are an important guide in decision making process.

Response: Yes, we agree with that. Since RPSs are rare tumors, the Trans-Atlantic RPS Working Group is now the most authoritative association on the management of RPS, and the updated Consensus on the management of primary RPS is now an important guide in decision making process.

The use of morbidity in the studies reviewed are more related with surgery complications than with long-term morbidity. A nice article that deals with this subject was published, I share the reference:

Severe chronic pain and lower limb motor impairment after multivisceral resection for retroperitoneal sarcomas are rare. Long-term renal function is not significantly impaired when nephrectomy is performed -- Callegaro D, Miceli R, Brunelli C, Colombo C, Sanfilippo R, Radaelli S, Casali PG, Caraceni A, Gronchi A, Fiore M. Long-term morbidity after multivisceral resection for retroperitoneal sarcoma. Br J Surg. 2015 Aug;102(9):1079-87. doi: 10.1002/bjs.9829. Epub 2015 Jun 3. PMID: 26041724.

Response: Thank you for your reminder. We have revised the use of morbidity to postoperative complication.

About the use of adjuvant radiotherapy in STS in the retroperitoneum the lack of high-level evidence supporting the benefit of these modality is crucial, but there are issues regarding acute and long-term toxicity not making this and absolute indication even when R2 resections are performed.

Response: Yes, we agree with that.

I agree that the indication and eligibility for surgical resection vary by medical center, but as you mentioned TARPSWG criteria give some backbones for decision making process, and as you say all patients must be referred to specialised centres because they are a significant challenge from a surgical standpoint.

Response: The first Consensus of TARPSWG on the management of RPS was proposed only 7 years ago. Many aspects of management of RPS need to be standardized, and that is why more studies need to be performed on this topic. 

When you mark the bias of this study I want to add

- Selection bias that is inevitably associated with this type of surgical studies

- Complexity to define R0 vs R1 in different retrospective and multi-institutional studies

- Do not take in account histological subtype and there different behaviours.

For example, well-differentiated liposarcoma essentially displayed a local risk of 20-40 % at 5 years after extended primary resection, and in some important series no patient developed metastatic disease.

On the other hand, leiomyosarcoma displayed a predominant systemic risk (greater than 40-50 %), with an extended primary approach optimized local control (95 % at 5 years in some series), but it could not prevent distant spread. Leiomyosarcoma had a better post-distant metastases outcome compared to other histologic subtypes.

Response: Yes, selection bias is inevitably associated with this type of surgical studies, especially when the indication and eligibility for surgical resection and the method of assessment of appropriate resection margins might vary by medical centre. We have revised the limitation part.

About your finals conclusions, I agree with them but not with the last one.

Meanwhile, unresectable recurrent RPS should be indicated for incomplete resection or debulking to improve survival -- I think that the indication for surgery after recurrence is very complex decision, and should be discussed on a multidisciplinary team individual cases considering histologic subtype and grade, disease free interval, presence of distant metastases, tumor multifocality, limited performance status, and complex multivisceral resections.

In the study of Grobney 48/61 with LR only underwent surgery.

Grochi wrote in the discussion that "Unlike the primary resection, the second surgery was not intended to remove adherent organs if not directly infiltrated, because there is no chance of cure after recurrence."

In the study of Lochan et al, 22/46 LR underwent surgery and there is no data about histologic type or grade.

ManNeill reported initial site of recurrence was local only (LR) for 219 patients, of whom 105 patients (48%) underwent surgery.

Response: Yes, we agree with that. Not all the unresectable recurrent RPS should be indicated for incomplete resection. We have revised the statement.

 

Reply to reviewer #3

Dear reviewer:

We appreciate your attitude to scientific review process and thank you for your comments. 

Reviewer #3: The background to the study was well laid and rationale. The presentation and structure of the abstract was however below par and could be improved. The manuscript was presented in standard English and made-for-easy reading. The intellectual content though satisfactory could be improved as suggested in the attached file.

Review

Survival Outcomes of Surgery for Retroperitoneal Sarcomas: A Systematic Review and Meta-analysis

Abstract – lacking some relevant sections. Study objectives were not stated. The abstract ought to necessarily state the inclusion/exclusion criteria for the articles used in the study. This was absent.

Response: The objectives were added in the revised abstract, and the selection criteria were also added in the revised abstract.

Introduction – a good background was laid for the study. The objectives are clearly stated and defined.

Methods – a good description of the inclusion/exclusion was provided. Search strategies were comprehensively explained. Attempts were made to determine the heterogeneity and the methodological qualities of the studies included.

Results – In the write up, the authors stated they included 47 articles in the meta-analysis. However, from the categorization, one will realize that there were 48 in all and not 47. 17 compared extended resection to tumor alone, 26 reported on surgical margins and 5 reported on long term outcomes.

Response: Of these 47 studies, one study (Bonvalot 2009, reference No. 20) not only compared extended resection to tumor alone, but also reported on surgical margins. 

Extended Resection vrs Tumor resection alone

The authors compared complications between extended resection and tumor resection alone and found no difference. It will be useful to know the complications common to both groups.

Response: Complication rates of each group were added in the results part.

Surgical resection margins

The section highlighted the importance of complete tumor resection. Patients with negative surgical margins had a better overall survival compared to those with positive surgeons. The distinction was even clearer when whose with R1 resection were compared to R2 resections. R1 resected tumors had a better overall survival than R2 tumors.

Even more, recurrent tumors managed by surgical intervention had a better long term survival compared to those managed conservatively.

Discussion

The discussion highlighted the importance of surgical resection, even in advance and recurrent tumors. It was demonstrated that overall survival even in R2 resected tumors were higher than those managed conservatively.

With regards to extended resection and the tumor resection alone groups, it would have been interesting to determine the rate of negative surgical margins in both groups since they seem to have similar morbidity, mortality and disease free survival rates. Is the surgical margin a determinant of the similar mortality and disease free rate? The authors could perform a sub-group analysis looking at this factor.

Response: Yes, we agree that surgical margin might be a determinant of the similar mortality and disease-free rate. After examine the included 17 studies compared between extended resection and tumour resection alone. We found that 7 studies mentioned the surgical margin of each group, and all the 7 studies owned similar surgical margin of each group. We performed sensitive analyses that only including studies with similar surgical margins. Interestingly, we found that the extended resection group had a significantly higher complication rate than the tumour resection alone group, which was not significantly higher in the initial analysis. We have added the results of sensitive analyses in the revised paper.

Conclusion

The conclusion was derived from the study findings and therefore appropriate

Other comments

A sub-group analysis looking at the role of radiotherapy and chemotherapy, and its influence on overall survival and disease free survival could have enriched the study findings. They could have also explored the role of neo-adjuvant vrs adjuvant therapy in the management of these cancers. 

Response: Since more than half of the included studies (26/47) did not mention the radiotherapy or chemotherapy, and the proportion of radiotherapy or chemotherapy of the studies varied, it is difficult to perform a sub-group analysis.

Recommendation

I recommend the publication of this manuscript subject to modifications based on the comments made above. The authors may choose not to respond to comments made under ‘other comments’.

Response: Thank you for offering your valuable comments.

---

## [Editor Report · Decision Letter 1]

13 Jul 2022

Survival Outcomes of Surgery for Retroperitoneal Sarcomas: A Systematic Review and Meta-analysis

PONE-D-21-35277R1

Dear Dr. Bin Huang,

We’re pleased to inform you that your manuscript has been judged scientifically suitable for publication and will be formally accepted for publication once it meets all outstanding technical requirements.

Kind regards,

Paolo Aurello

Academic Editor

PLOS ONE
---

## [Editor Report · Acceptance letter]

19 Jul 2022

PONE-D-21-35277R1 

Survival Outcomes of Surgery for Retroperitoneal Sarcomas: A Systematic Review and Meta-analysis 

Dear Dr. Huang:

I'm pleased to inform you that your manuscript has been deemed suitable for publication in PLOS ONE. Congratulations! Your manuscript is now with our production department. 

Kind regards, 

on behalf of

Dr. Paolo Aurello 

Academic Editor

PLOS ONE